# Floquet parity-time symmetry in integrated photonics

Weijie Liu [1,7], Quancheng Liu [2,7], Xiang Ni[3,4,7], Yuechen Jia [1], Klaus Ziegler [5], Andrea Alù [4,6] ✉ & Feng Chen [1] ✉

Parity-time (PT) symmetry has been unveiling new photonic regimes in non-Hermitian systems, with opportunities for lasing, sensing and enhanced light-matter interactions. The most exotic responses emerge at the exceptional point (EP) and in the broken PT-symmetry phase, yet in conventional PT-symmetric systems these regimes require large levels of gain and loss, posing remarkable challenges in practical settings. Floquet PT-symmetry, which may be realized by periodically flipping the effective gain/loss distribution in time, can relax these requirements and tailor the EP and PT-symmetry phases through the modulation period. Here, we explore Floquet PT-symmetry in an integrated photonic waveguide platform, in which the role of time is replaced by the propagation direction. We experimentally demonstrate spontaneous PT-symmetry breaking at small gain/loss levels and efficient control of amplification and suppression through the excitation ports. Our work introduces the advantages of Floquet PT-symmetry in a practical integrated photonic setting, enabling a powerful platform to observe PT-symmetric phenomena and leverage their extreme features, with applications in nanophotonics, coherent control of nanoscale light amplification and routing.

Engineering the interplay between gain and loss is at the basis of parity-time (PT) symmetry[1–5], which has recently received extensive attention, both for fundamental research[6–10] and advanced technologies[11–19]. The concept of PT-symmetry has originated from the study of open quantum systems, which are invariant under combined parity-time symmetry transformations, and can exhibit a real-valued energy spectrum described by non-Hermitian quantum theory[20–22]. PT-symmetry has found fertile grounds in classical wave physics, from photonics to acoustics, triggering the exploration of various counterintuitive phenomena, such as anyonic-PT-symmetry controlled lasing[23], unidirectional invisibility[24], negative refraction[25], power oscillations[26] and nonreciprocal transmission[27]. Several experimental platforms, including electronics[28], acoustics[29], and matter waves[30], have been proposed and implemented to showcase the wave phenomena enabled by PT-symmetry in classical wave systems. In particular, the analogy between the paraxial Helmholtz wave equation of coupled waveguide systems in space and the Schrödinger equation in the time domain has inspired various setups capable of practically demonstrating PT-symmetric phenomena both in linear and nonlinear regimes[31–39].

Conventional PT-symmetric systems require balanced distributions in the space of gain and loss. Alternatively, passive PT-symmetric systems can be implemented in photonic waveguides by adding a uniform loss offset, and spatially tailoring the loss in a balanced way, e.g., using curved waveguides[40,41]. These strategies of adding loss have been implemented to control phase transitions between broken and unbroken PT-symmetry phases without requiring gain.

---

[1]School of Physics, State Key Laboratory of Crystal Materials, Shandong University, 250100 Jinan, China. [2]Department of Physics, Institute of Nanotechnology and Advanced Materials, Bar-Ilan University, Ramat-Gan 52900, Israel. [3]School of Physics, Central South University, 410083 Changsha, Hunan, China. [4]Photonics Initiative, Advanced Science Research Center, City University of New York, New York, NY 10031, USA. [5]Institut für Physik, Universität Augsburg, 86135 Augsburg, Germany. [6]Physics Program, Graduate Center, City University of New York, New York, NY 10016, USA. [7]These authors contributed equally: Weijie Liu, Quancheng Liu, Xiang Ni. ✉e-mail: aalu@gc.cuny.edu; drfchen@sdu.edu.cn

Periodic driving of PT-symmetric systems breaks time translational symmetry and allows for enriched phase transitions[42–45]. For example, periodic modulation of the non-Hermiticity parameter was introduced in a two-level Rabi model, leading to a broken PT-symmetry phase at arbitrary levels of gain/loss[46]. On the other hand, non-Hermitian systems subject to periodic modulation, without possessing PT symmetry, have the potential to enter the pseudo-PT-symmetry phase[47] or achieve an effective real spectrum[48]. Additionally, Floquet PT-symmetric systems with time-periodic dissipation and couplings have been realized in active circuit resonators[49] and ultracold atoms[18], demonstrating PT-symmetry phase transitions at small dissipation regimes. Recently, a novel topological phase of the Floquet PT-symmetric SSH model was discovered, in which the Floquet modulation is realized through the modulation of hopping terms in the SSH lattice[50]. However, these platforms either have used active elements or need intricate setups to achieve the desired periodic driving.

In this work, we leverage periodic spatial modulation of loss in an integrated waveguide configuration to realize Floquet PT-symmetry in integrated photonics. We unveil unprecedented control of phase transitions and effective amplification regimes based on Floquet non-Hermitian photonics. By properly tailoring the Floquet period, we demonstrate PT-symmetry phase transitions with adjustable levels of gain/loss. Remarkably, in our platform we also demonstrate extreme control of the response of the system through the excitation ports, switching suppression/amplification regimes in real time. Our study opens exciting possibilities for advanced manipulation and control of light propagation based on Floquet PT-symmetry, with exciting applications in integrated photonics and nanophotonics.

## Results

In non-Hermitian PT-symmetric systems, we can define a *directionality*, stemming from the broken parity-symmetry. For instance, a PT-symmetric structure with loss in the bottom and gain in the top can be defined as *up*, while *down* indicates gain in the bottom and loss in the top. Hence, the PT-symmetric Hamiltonians $H_{pt}^\uparrow$ and $H_{pt}^\downarrow$ correspond to loss/gain and gain/loss, respectively (Fig. 1a)

$$H_{pt}^\uparrow = \begin{pmatrix} i\Gamma & \kappa \\ \kappa & -i\Gamma \end{pmatrix}, H_{pt}^\downarrow = \begin{pmatrix} -i\Gamma & \kappa \\ \kappa & i\Gamma \end{pmatrix}, \tag{1}$$

where $\Gamma$ denotes the non-Hermiticity parameter and $\kappa$ the coupling strength. Both these Hamiltonians are invariant under the PT symmetry operator, $PTH_{pt}^\uparrow PT = H_{pt}^\uparrow$ and $PTH_{pt}^\downarrow PT = H_{pt}^\downarrow$, where $P = \begin{pmatrix} 0 & 1 \\ 1 & 0 \end{pmatrix}$ denotes spatial inversion, and $T$, represented by complex conjugation, reverses the direction of time. In a static PT-symmetric system consisting of $H_{pt}^\uparrow$ or $H_{pt}^\downarrow$, the response is in the PT-symmetry phase when $\Gamma < \kappa$, i.e., for small gain/loss, and its amplitude remains constant in time. In the broken phase $\Gamma < \kappa$, one of the modes is amplified exponentially, while the other decays.

In our setting, the Floquet PT-symmetric system is designed by periodically swapping the locations of loss and gain in time (Fig. 1b), while preserving PT-symmetry at each time instant. We note there are alternative definitions of Floquet PT symmetry in the literature[18,24,50], which are also valid. Our system is represented by $H_{pt}^\uparrow$ for time $\tau_\uparrow$ and then $H_{pt}^\downarrow$ for time $\tau_\downarrow$, i.e., $H(t + T) = H(t)$ where $T = \tau_\uparrow + \tau_\downarrow$ (Fig. 1c). The period controls output of the system in nontrivial ways. The Hamiltonian $H(t)$ describing this Floquet process is PT-symmetric, i.e., $PTH(t)PT = H(t)$, while its properties are remarkably different compared to the two static PT-symmetric systems forming its evolution, as we discuss in the following. We observe that $H(t)$ exhibits PT symmetry at all instants in time, regardless of the initial condition (whether $t = 0$ or not) of the system. We experimentally demonstrate our findings in an integrated photonic platform by replacing time with the propagation direction in a periodically varying evanescently coupled waveguide array consisting of alternative straight (lower loss) and tailored curved (higher loss) waveguide sections (Fig. 1d).

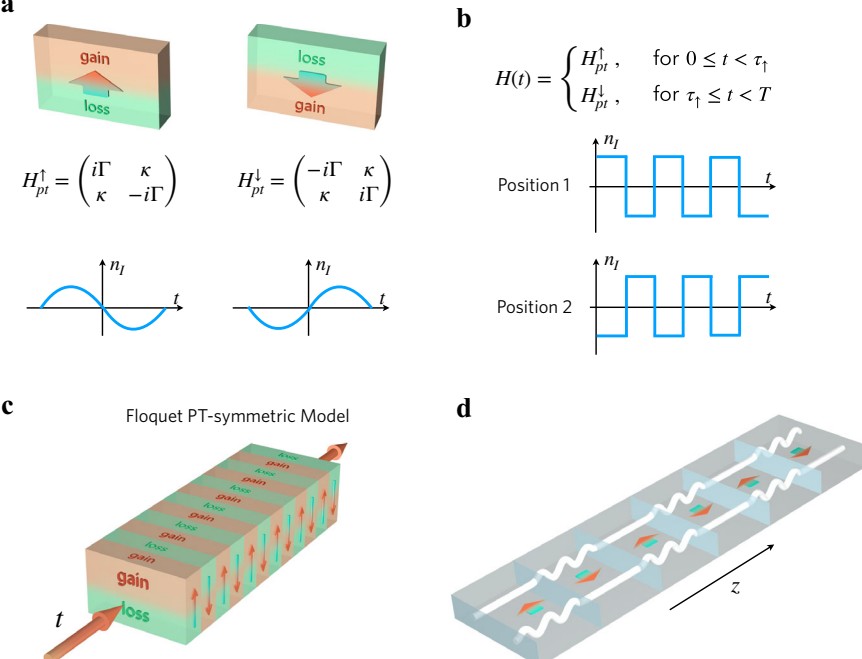

**Fig. 1 | Schematic plot of a PT-symmetric model and its analog in integrated photonics. a** Definition of the PT-symmetric Hamiltonians $H_{pt}^\uparrow$ and $H_{pt}^\downarrow$. **b** Time-dependent Hamiltonian $H(t)$ consisting of periodically swapped up and down PT-symmetric Hamiltonians. **c** Schematic plot of the PT-symmetric system with periodic modulation. **d** Schematic plot of the waveguide arrays for the experimental realization of Floquet PT-symmetric systems. Here the straight waveguide forms the gain portion, and the curved waveguide realizes the lossy portion.

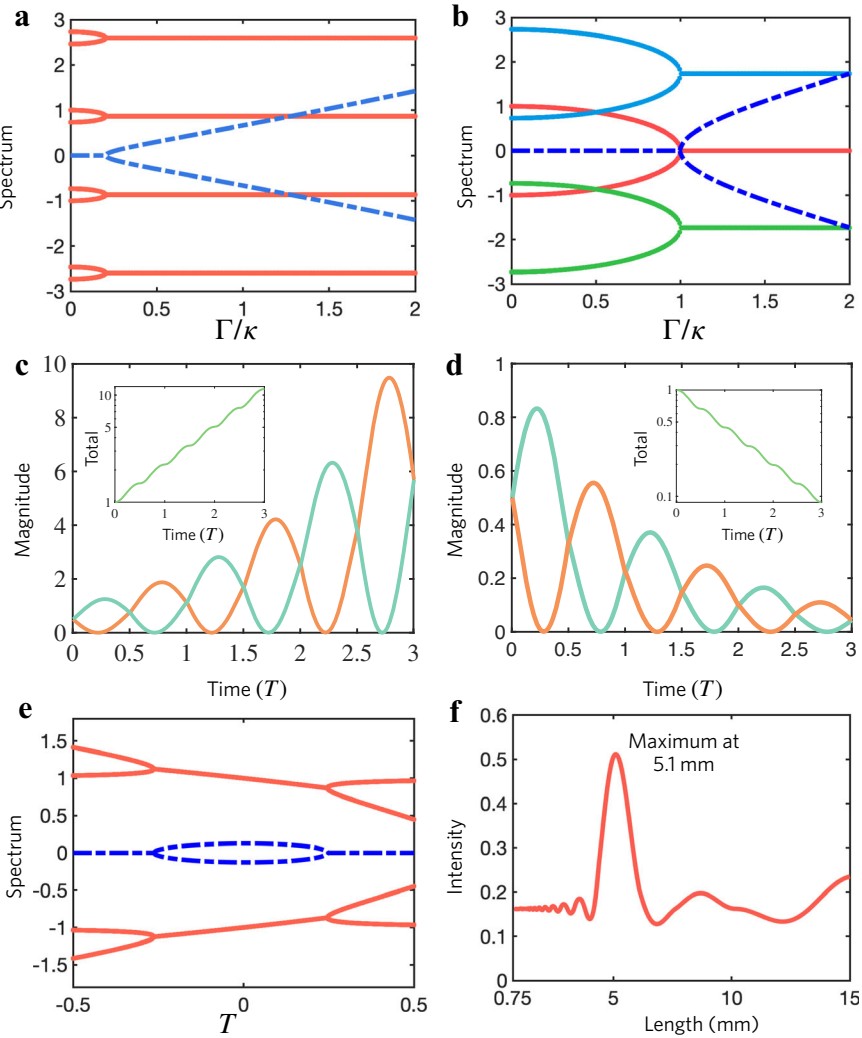

**Fig. 2 | Floquet quasi-energy spectrum, PT-symmetry phase transition and dynamic evolution of the Floquet PT-symmetric system. a** Imaginary (blue) and real (red) parts of the quasi-energy spectrum of the effective Hamiltonian $H_{eff}$ for the Floquet PT-symmetric systems. where the breaking of PT-symmetry appears at the EP with small $\Gamma$. **b** Real part energy spectra and its translation in frequency for ordinary PT-symmetric system. The dashed line denotes its imaginary part of the energy spectrum. **c, d** Evolution of the amplification $|\xi_{am}\rangle$ and suppression $|\xi_{su}\rangle$

modes of the two outputs of the PT-symmetric system (total intensity in the inset). They have the same magnitude in two locations but different phases on the Bloch sphere. Here $\Gamma = 0.2$ and $T = 5\pi/(2\sqrt{6})$ (see details in SI-3(S10)). **e** Imaginary (blue) and real (red) quasi-energy of Floquet PT-symmetric system versus periodicity $T\kappa$. **f** Simulated (based on beam-propagation method) light output intensity versus length of Floquet periodicity in the waveguide array.

Following Floquet theory, the stroboscopic dynamics of our Floquet Hamiltonian are described by the time-evolution operator over a complete driving period, i.e., $U^{\uparrow,\downarrow} = \mathcal{T} e^{-i\int_0^T H(t)dt}$, where $\mathcal{T}$ executes the time ordering, and $T$ is the temporal period. We calculate the effective Hamiltonian $H_{eff}$ of the Floquet PT model given by $H_{eff} = i\log[U^{\uparrow,\downarrow}(T)]/T$. In Fig. 2a, we show the quasi-energy spectra (real part, red; imaginary part, blue) for $T = 3.626/\kappa$, compared to the spectrum in Fig. 2b (red curve, red; imaginary part, blue), corresponding to the static scenario. The Floquet bands repeat themselves in quasifrequency dimension because the discrete time translation symmetry is preserved up to the period $T$. It is evident that the PT-symmetry phase transition, in which the spectrum evolves from real-valued to complex-valued through exceptional points (EPs), occurs at much smaller values of $\Gamma$ for the Floquet system compared to the static regime. To explain this finding, we translate the static PT-symmetric energy spectrum (red line) by the Floquet frequency $\Omega = 2\pi/T$, resulting in band crossing of different Floquet orders. The interaction of these Floquet bands under the periodic modulation leads to the

emergence of new EPs, and PT-symmetry phase transitions in the Floquet quasi-energy spectra of Fig. 2a. Hence, the crossing points and resulting EPs can be tuned through the Floquet frequency $\Omega$, and the Floquet PT-symmetric system can be in the broken phase even for vanishingly small non-Hermiticity parameter, provided that the Floquet frequency matches the bandwidth of Hermitian energy $\Omega \approx 2\kappa$. Overall, the Floquet PT-symmetric system enables wide control over EPs and phase transitions through the modulation period, facilitating the implementation of exotic wave phenomena.

When entering the broken phase, coherent control of the amplification and decay of the input signals can be efficiently realized. The time-evolution of the Floquet system can be formally written in terms of the time-evolution operator $|\psi(nT)\rangle = (U^{\uparrow,\downarrow})^n|\psi_0\rangle$, with $|\psi_0\rangle$ being the initial state of the system and $U^{\uparrow,\downarrow}$ being the eigenmodes of the time-evolution operator. We study a special example in which the effective Hamiltonian can be analytically studied (see details in Supplementary S3) and the eigenstates of $U^{\uparrow,\downarrow}$ are $|\xi_{am}\rangle = \{1/\sqrt{2}, -i/\sqrt{2}\}$ and $|\xi_{su}\rangle = \{-i/\sqrt{2}, 1/\sqrt{2}\}$, with eigenvalues $\xi_{am} = (1+\Gamma)/(-1+\Gamma)$ and

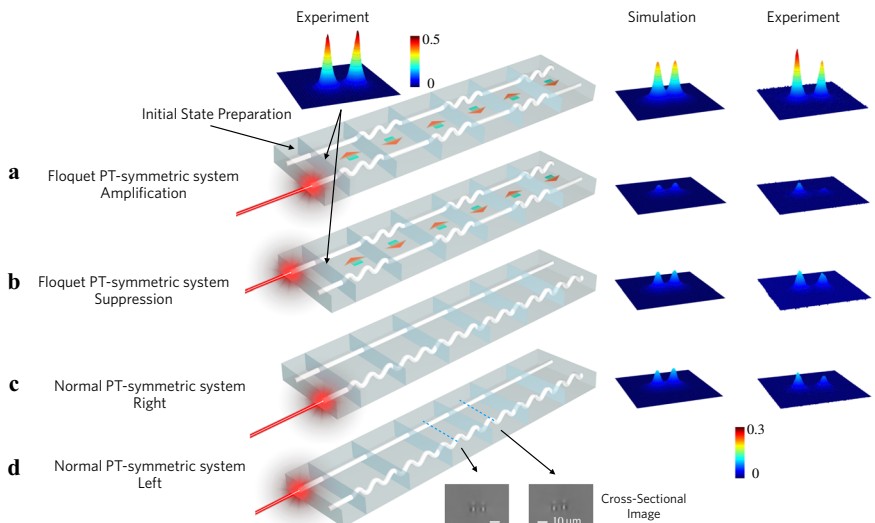

**Fig. 3 | Experimental realization of Floquet PT-symmetric system and static PT-symmetric system. a, b** Straight waveguides of 2.5 mm mutual coupling length are employed to prepare initial states $|\xi_{am}\rangle$ and $|\xi_{su}\rangle$. The panel above the first structure shows the light intensity distribution of the initial state. In figure **a, b**, both straight and curved waveguides are periodically flipped, each unit bearing a length $L = T/2 = 5.1$ mm. The array is independently excited from the right/left waveguide for amplification/suppression modes. Light traverses the PT-symmetric system for three periods, covering a total length $L_{total} = 3\,T = 30.6$ mm, with light distribution gauged at the array's terminus. The Beam-Propagation Method is utilized to simulate light evolution, with corresponding simulated and experimental outcomes displayed on the right. **c, d** Waveguide arrays and light intensity for static PT-symmetric cases. Light is introduced from the right in figure **c** and from the left in figure **d**, each corresponding to the cases displayed in figure **a, b**, respectively. The panel below the last structure shows the experimental micrograph of the facets of the waveguide arrays.

$\xi_{su} = (-1 + \Gamma)/(1 + \Gamma)$, corresponding to amplified and suppressed states, respectively. These states have a $\pi/2$ difference in phase on the Bloch sphere of a two-dimensional Hilbert space. When $|\psi_0\rangle = |\xi_{am}\rangle$, the evolution of the system reads $|\psi(nT_{evol})\rangle = (U^{\uparrow,\downarrow})^n|\xi_{am}\rangle = \xi_{am}^n|\xi_{am}\rangle$ and the total intensity at the output is $I(nT_{evol}) = \xi_{am}^{2n}I_0$ (Fig. 2c). Since $\xi_{am} > 1$, the PT-symmetric system exponentially amplifies this mode, and the amplification magnitude can be precisely controlled by the parameter $\Gamma$. On the contrary, when $|\psi_0\rangle = |\xi_{su}\rangle$, we have $|\psi(nT_{evol})\rangle = (U^{\uparrow,\downarrow})^n|\xi_{su}\rangle = \xi_{su}^n|\xi_{su}\rangle$ and $I(nT_{evol}) = \xi_{su}^{2n}I_0$, due to the fact that $\xi_{su} < 1$, such initial state is exponentially dampened by the PT-symmetric system (Fig. 2d).

Interestingly, by tuning the Floquet periodicity of our system we can tailor the signal amplification for given non-Hermiticity parameter $\Gamma$. For example, choosing the gain/loss parameter $\Gamma = 0.2\kappa$, we calculate the quasi-energy of the Floquet PT-symmetric system as a function of the temporal periodicity. As shown in Fig. 2e, the Floquet period can be tailored to enter the broken PT-symmetric phase, and for $T = 3.15/\kappa$ maximum imaginary quasi-energy is obtained, implying maximum amplification/decay of the signal at the output.

To verify our theory, we performed beam-propagation simulations in Floquet waveguide arrays, in which the temporal dimension is replaced by the direction of propagation. By varying the winding as in Fig. 1d, we can effectively build a Floquet PT-symmetric system, and we can tailor the spatial modulation period to control the output at the end facets of the waveguide array (Supplementary S1, Fig. S1). The weakly guided propagation (along $z$) of optical waves in such a waveguide array follows the Schrödinger-type paraxial wave equation $-i\partial\psi(x,y;z)/\partial z = [\nabla_\perp^2/2k + \Delta n(x,y;z)k/n_0]\psi(x,y;z) = H\psi(x,y;z)$, where $\nabla_\perp^2 = \partial^2/\partial^2 x + \partial^2/\partial^2 y$, $\psi$ is the envelope of the light field of wavelength $\lambda$, corresponding to the wavenumber $k = 2\pi/\lambda$, $n_0$ is the substrate refractive index, and $\Delta n(x, y; z)$ is the index modulation[51–54]. This equation can be mapped to the conventional Schrödinger equation for $z \rightarrow t$ and $-\Delta n \rightarrow V$. Hence, the time-evolution of a Hamiltonian system in Eq. (1) can be mapped to optical wave propagation along $z$ in the waveguide. When light propagates in the curved waveguide section, additional bending loss is experienced. This process effectively introduces an imaginary term to the optical potential in the paraxial Helmholtz

equation. We calibrate the decay rate $\Gamma_{sw}$ of the curved waveguide sections with independent experiments (see details in Supplementary S2 and Fig. S2). By offsetting an additional common loss term, present in both the straight and curved sections and given by the passive nature of the system $-i\Gamma_{sw}\sigma_0/2 = -i\Gamma\sigma_0$ ($\sigma_0$ is the $2 \times 2$ identity matrix)[55,56], our platform realizes a passive PT-symmetric Hamiltonian and allow precise control over the Floquet period. The straight sections with no additional loss correspond to effective gain regions, and the curved sections with additional loss $\Gamma_{sw}$ to lossy regions, and the non-Hermiticity parameter of the system is the $\Gamma_{sw}/2$. The parameters are chosen such that $\Gamma < \kappa$, hence each static PT-symmetric Hamiltonian is in the PT-symmetry phase, for which the energy spectrum would be real. By periodically swapping the effective gain and lossy region in $z$ with periodicity $L$, we realize a passive Floquet PT-symmetric system emulating our theoretical model. As the length $L$ changes from 4 mm to 6.75 mm, the system enters the broken PT-symmetry regime, and we obtain maximum output at $L = 5.1$ mm, as shown in Fig. 2f, indicating maximum signal amplification in the photonic platform, details in Fig. S3. When L is less than 4 mm or larger than 6.5 mm, the system is in the PT-symmetry regime and it exhibits oscillatory behavior as the periodicity varies, because of its real eigenspectra. These simulations in both broken and unbroken PT-symmetry regimes are consistent with our theoretical results and validate the mapping of our model into an integrated photonic platform.

To experimentally demonstrate Floquet PT-symmetry and its remarkable features, the preparation of excitation states $|\xi_{am}\rangle$ and $|\xi_{su}\rangle$ is important. We can use two coupled straight waveguides (labeled by WG1 and WG2) to feed the Floquet PT-symmetric lattice shown in Fig. 3, with length $L = T_R/4$, where $T_R$ denotes the Rabi oscillation period of the coupled straight waveguides (Fig. 3). Using a 633-nm He-Ne laser to excite the system from WG1, the state of light, after a propagation length $T_R/4$, is the amplified initial state, i.e., $|\psi(0.25T_R)_{WG1}\rangle = |\xi_{am}\rangle$ (Fig. 3a). When we excite the system from WG2, the system attains the state $|\psi(0.25T_R)_{WG2}\rangle = |\xi_{su}\rangle$ after a length $T_R/4$ (Fig. 3b). In both instances, straight waveguides of 2.5 mm mutual coupling length are employed to make sure that the initial states $|\xi_{am}\rangle$ and $|\xi_{su}\rangle$ impinge on the PT-symmetric integrated photonics for

excitation of one of the ports. Subsequently, the light propagates in the Floquet PT-symmetric system, with a waveguide separation of $d = 10\,\mu m$. According to the simulation, the Floquet periodicity length is chosen as $T = 10.2\,mm$ such that the maximum amplification/ decaying of broken PT-symmetry phase is achieved, with $\kappa = 314\,m^{-1}$, and $\Gamma = 50.2\,m^{-1}$ (SI, Fig. 2). The output modal profiles correlated with light propagation are then monitored. Figure 3c, d demonstrate the static PT-symmetric cases, where one is a straight (low-loss) waveguide, and the other is curved (high-loss). Straight waveguide arrays are still employed ahead of the static PT-symmetric system to generate the two initial states.

As depicted in Fig. 3, the geometries of four different waveguide arrays are shown in the left panel. The experimental and simulated light distribution at the output of the array for the Floquet PT-symmetric system and static PT-symmetric system are plotted in the middle and right panels, respectively. Despite both systems

starting from identical light intensity, the light intensity in the amplification regime of Floquet PT-symmetric photonics vastly surpasses one of the static PT-symmetric systems. Light intensity in the suppression regime decays more rapidly compared to the static PT-symmetric case. These experimental findings are consistent with theoretical predictions, validating the intriguing features of Floquet PT-symmetric systems in tailoring the PT-symmetry phase transition and maximizing the amplification/decay regime for a given level of non-Hermiticity parameter.

Figure 4a demonstrates experimental (right panel) and simulated (left panel) intensity distributions versus propagation distance $z$ for the Floquet PT-symmetric system, showing amplification and suppression modes, and the static PT-symmetric scenarios for comparison, consistent with Fig. 3. As the light propagates through the waveguide array, the light intensity in the amplification mode substantially exceeds the one observed in the static PT-symmetric geometries, whereas the light intensity in the static PT-symmetric system surpasses the one of the suppressed modes. The experimental data aligns well with the simulations. Figure 4b shows the theoretical (depicted by lines, utilizing Eq. (1)) and experimental (portrayed by dots) demonstration of light intensity within the system versus propagation distance. The light intensity for the amplification, suppression modes, and the static PT-symmetric system are shown. As illustrated in the figure, it can be observed that the amplification mode of the light intensity surpasses that of the static PT-symmetric system scenarios and the suppression mode. The suppression mode exhibits a decay rate swifter than the static PT-symmetric system. These findings underscore the control over PT-symmetry breaking through Floquet mechanisms, thereby empowering control over both amplification and suppression, even in the case of small gain/loss regimes.

## Discussion

By periodically swapping non-Hermitian PT-symmetric Hamiltonians, we introduced a Floquet PT-symmetric model and utilized the Floquet mechanism to control PT-symmetry phase transitions and exceptional points, as well as the associated complex spectra and dynamics. Our analytical and experimental results demonstrate the rich physics of Floquet PT-symmetric systems can be applied in an integrated photonic platform, featuring a spontaneous broken PT-symmetry phase through the Floquet modulation. Intriguingly, our Floquet PT-symmetric model possesses the remarkable ability to exponentially amplify or suppress input signals, offering a unique means to control light output in artificial photonic structures and unlocking exciting possibilities for the development of on-chip devices for light control (see Supplementary S4 for an example, control by phases and loss magnitudes are shown in Figs. S4 and S5, respectively). Furthermore, by carefully selecting the periodicity of the Floquet system, we have demonstrated maximum signal amplification at the output of our photonic structures. Our work paves the way for larger control over Floquet PT-symmetric systems and enables functional and efficient manipulation of light transport for potential applications in integrated photonics and nanophotonics.

## Methods

### Fabrication and characterization

The Floquet PT-symmetric waveguide structures used in this work are fabricated in commercially available borosilicate glass (Eagle XG) using the well-developed technique of direct femtosecond-laser writing. The glass is mounted on a computer-controlled 3D $x$-$y$-$z$ translation stage (Hybrid Hexapod, ALIO). A femtosecond laser (Femto YL-25, YSL Photonics) at a wavelength of 1030 nm, a pulse duration of 400 fs, and a repetition rate of 2.5 MHz is used as the light source. A microscope objective (50×/0.45 N.A.) is utilized to focus the laser beam at ~230 µm below the substrate surface of the sample. The pulse energy is adjusted to ~212 nJ, and the writing speed is set to 1 mm·s⁻¹. The laser writing

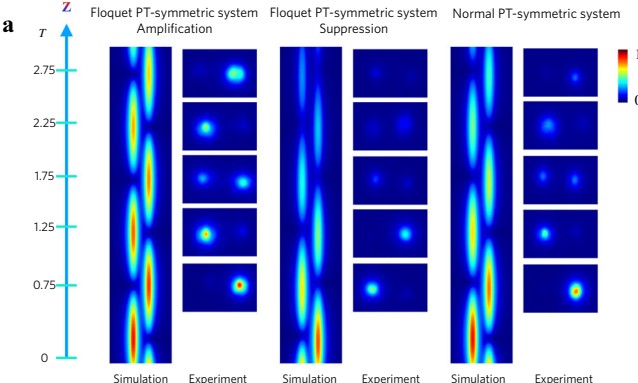

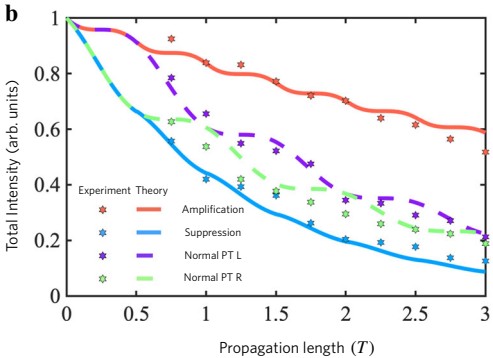

**Fig. 4 | Measured light intensity for Floquet PT-symmetric system and static PT-symmetric system. a** Experimental (left) and simulated (right, employing the beam propagation method) light distribution versus propagation distance $z$ for the Floquet PT-symmetric system, featuring amplification and suppression modes, and conventional PT-symmetric systems. As light propagates through the waveguide array, the light intensity in the amplification mode considerably surpasses the one observed in static PT-symmetric system scenarios, whereas the regular PT system surpasses the suppression mode. This experimental data is in agreement with the simulations, corroborating the unveiled physics of the Floquet setup. **b** Theoretical (represented by lines, with Eq. (1)) and experimental (depicted by dots) distributions of total light intensity within the system versus propagation distance. The light intensity for the amplification, suppression modes, and also the static PT-symmetric case are measured. As illustrated in the figure, the light intensity of the amplification mode significantly surpasses the one in static PT-symmetric systems and the suppression mode. The suppression mode exhibits a decay rate faster than the one of the static PT-symmetric system. These findings demonstrate that the Floquet PT-symmetric system spontaneously breaks the PT-symmetry phase in a regime where the static PT-symmetric system preserves the PT-symmetry phase, thus empowering control over both amplification and suppression, even amidst minimal loss and gain.

produces typical Type I waveguides with a width of ~ 4 μm, and the refractive index difference Δ*n* between waveguide cores and cladding is ~10⁻³. The characterization of the waveguide system is performed by an end-face coupling system. The coupling strengths are determined by the spacings of the adjacent waveguides. A He-Ne laser (at a wavelength of 633 nm) is injected into the selected waveguide of the system by a microscope objective (20×/0.4 N.A.) to excite waveguide modes. The output intensity distribution (beam evolution) is measured (monitored) at the output facet using a CCD camera by another microscope objective (20×/0.4 N.A.). The waveguide losses are determined to be less than 0.5 dB/cm. The theoretical approach and numerical method can be seen in Supplementary section S3.

## Data availability

Relevant data supporting the key findings of this study are available within the article, the Supplementary Information file, and the Source data file. All the data generated in this study have been deposited in the Figshare database under [https://doi.org/10.6084/m9.figshare.24981819]. Source data are provided in this paper.

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

## Acknowledgements

This work was supported by the National Natural Science Foundation of China (Grant No. 12174222), Natural Science Foundation of Shandong Province (Grant No. ZR2021ZD02), and Taishan Scholars Project of Shandong Province (tspd20210303). Q.L. acknowledges support from the Israel Science Foundation (Grant No. 1614/21). K.Z. acknowledges support from the Julian Schwinger Foundation for Physics Research. X.N. and A.A. were supported by the Simons Foundation and the Air Force Office of Scientific Research MURI program. X.N. acknowledges support from Research Startup Funds of Central South University (Grant No. 11400-506030109).

## Author contributions

F.C. and A.A. proposed the concept and the modeling. Q.L., K.Z., and X.N. contributed to the theoretical analysis and designed the setting up of the experiments together with W.L. W.L., Y.J., and F.C. performed the experiment and analyzed the data together with Q.L. and X.N. F.C. and A.A. supervised the project. All authors contributed to the preparation of the manuscript.

## Competing interests

The authors declare no competing interests.
