## [Peer Review File · Nature Communications]

REVIEWER COMMENTS

Reviewer #1 (Remarks to the Author):

The authors of this manuscript report the first optical experiment on Floquet PT-symmetric systems. They have used an isomorphism between the paraxial equation (applicable for waveguide platforms) and the time-modulated Schroedinger equation in order to analyze the Floquet spectrum and associated signal propagation. The experimental demonstration is novel and the theoretical analysis is scientifically sound and in agreement with the experimental findings. I recommend the paper for publication to Nature Communications since it is the first optical demonstration of such a setting and might potentially inspire some photonics technology. At the same time, I have a number of comments that the authors have to address:

1) There is a recent publication that actually analyzes the noise effects in the case of EP sensing that utilizes a transport protocol: Exceptional-point-based accelerometers with enhanced signal-to-noise ratio, R. Kononchuk, J. Cai, F. Ellis, R. Thevamaran, T. Kottos, nature 607, 697 (2022). I think that the authors have to refer to this at the introduction where they discuss about "advanced technologies".

2) Towards the end of the introductory paragraph the authors discuss about "Several experimental platforms..." that demonstrate PT-symmetry. Indeed there are electronics, acoustic, matter waves, spintronics ect. The authors have to give some representation references there.

3) It is probably a "language" slip but at the top of page 3 (second paragraph before the end of introduction) the authors state that the previously used Floquet platforms "require active elements". I do not think that they are "requiring" active elements. But it is true that they have been performed using active elements. I think that it is more correct to say instead "...have used active elements...".

4) At the section of "Experimental realization of the Floquet PT-symmetric System" there are many typos (most coming from the "ket" notations. Please correct.

5) In the case of the amplified signal, are there any nonlinear saturation effects? If yes, is there an expectation when they kick in?

6)Reference 34: There is a missing "PT" in the title of this paper. Also in Ref. 34 the title starts with small letter i.e. "spectroscopy...". Please correct. Also, the authors might want to cite the following theoretical work that also discuss Floquet PT-symmetric systems (although this is not necessary):

Wave-packet self-imaging and giant recombinations via stable Bloch-Zener oscillations in photonics lattices with local PT-symmetry, N. Bender, H. Li, F. M. Ellis, T. Kottos, Phys. Rev. A 92, 041803(R) (2015).

Provided that the above comments are addressed appropriately, I recommend publication of this paper to Nature Communications.

Reviewer #2 (Remarks to the Author):

This manuscript presents an experimental realisation of an integrated optical waveguide coupler with specially introduced periodically modulated loss, such that the evolution is mathematically equivalent to a Floquet parity-time (PT) symmetry dynamics. While this problem has been extensively investigated theoretically and experimentally in other contexts, the presented realisation with integrated waveguides appears to be novel.

There appear several questions

1) It is stated in lines 93-94 that "The Hamiltonian $H(t)$ describing this Floquet process is PT-symmetric, i.e., $PTH(t)PT=H(t)$ ". However, this is not generally correct, depending on where the zero of the time (distance) axis is selected. Specifically, only if $t=0$ is in the *_middle_* of one segment with loss in either left or right waveguides, then indeed $PTH(t)PT=H(t)$. But if $t=0$ is at the boundary between the "up" and "down" segments, then one actually identifies *_anti_* PT symmetry since the time-reversal operator swaps the two segment types, meaning that there appears a minus sign on the RHS as $PTH(t)PT=-H(t)$. Please clarify and add a critical discussion on these aspects.

2) It would be useful to cite and discuss the following relevant references: Pseudo-Parity-Time Symmetry in Optical Systems <https://doi.org/10.1103/PhysRevLett.110.243902>, Stabilizing non-Hermitian systems by periodic driving <https://doi.org/10.1103/PhysRevA.91.042135>

Overall, this work can be potentially recommended for publication after a revision.

Reviewer #3 (Remarks to the Author):

In this work, the authors designed an integrated photonic waveguide and explored the Floquet PT-symmetry. They experimentally demonstrated the PT-symmetry breaking transition and achieved the control of amplification and absorption through excitation ports in their setup.

Overall, this paper is clearly written. The reported results could make an interesting contribution to the experimental realization of non-Hermitian Floquet systems. However, the PT symmetry, exceptional point and the related non-Hermitian phenomena have been intensively studied in the past decades. In photonic waveguides, the PT symmetry and its breaking have been considered in many theoretical and experimental studies, e.g., see the following research articles and reviews:

PHYSICAL REVIEW RESEARCH 3, 023211 (2021)

<https://doi.org/10.1002/adma.201903639>

<https://doi.org/10.1002/apxr.202200053>

<https://doi.org/10.1093/nsr/nwy011>

Phys. Rev. Lett. 101, 080402

Physics - Uspekhi 57 (11) 1063 - 1082 (2014)

<https://doi.org/10.1364/PRJ.6.000A51>

10.1126/sciadv.abo6220

Specially, an experimental realization of a non-Hermitian Floquet system was reported before in PHYSICAL REVIEW RESEARCH 3, 023211 (2021). Replacing the role of time by a spatial propagation direction is also a well-developed approach to do Floquet engineering in photonic waveguides. Meanwhile, there are many other theoretical works on PT symmetric and more generally non-Hermitian Floquet systems, which are unfortunately ignored by the authors. Putting together, I think that regarding the impact and novelty, the results reported in this work could not meet the high standards of nature communications. Therefore, I could not recommend publication.

There are some other issues that I suggest the authors to address before they resubmit this paper to any other journals:

(i) Some descriptions and definitions in the paper are not accurate. For example, having balanced gain and loss is not the necessary condition for the appearance of PT-broken phases and exceptional points.

Spatially asymmetric/nonreciprocal couplings or even a uniform loss could do the job. The authors' related statements in the abstract are thus inaccurate.

(ii) The concept of Floquet PT-symmetry should be treated more carefully. For stroboscopic dynamics, the Floquet PT-symmetry should be defined at the level of Floquet operator. In more general situations, one should consider the impact of PT operations on both the effective Hamiltonian and micromotion operator. Defining the Floquet PT-symmetry simply as the PT symmetry of the instantaneous time-dependent Hamiltonian is questionable.

(iii) Some obvious typos should be removed. In the lines 181-189 of the PDF file, there are some strange symbols, making the text unreadable. In Refs. [33] and [34], the PT symbols are missing. The authors are suggested to check their text more carefully.

Response Letter

Responses to Reviewer #1

The authors of this manuscript report the first optical experiment on Floquet PT-symmetric systems. They have used an isomorphism between the paraxial equation (applicable for waveguide platforms) and the time-modulated Schrodinger equation in order to analyze the Floquet spectrum and associated signal propagation. The experimental demonstration is novel and the theoretical analysis is scientifically sound and in agreement with the experimental findings. I recommend the paper for publication to Nature Communications since it is the first optical demonstration of such a setting and might potentially inspire some photonics technology. At the same time, I have a number of comments that the authors have to address:

Reply: We thank the Reviewer #1 for highlighting the novelty of our manuscript and for the recommendation for publication in Nature Communications.

1) There is a recent publication that actually analyzes the noise effects in the case of EP sensing that utilizes a transport protocol: Exceptional-point-based accelerometers with enhanced signal-to-noise ratio, R. Kononchuk, J. Cai, F. Ellis, R. Thevamaran, T. Kottos, nature 607, 697 (2022). I think that the authors have to refer to this at the introduction where they discuss about "advanced technologies".

Reply: We thank the Reviewer #1 for bringing this work to our attention, which utilizes exceptional-point-based PT-symmetric electromechanical accelerometers to enhance signal-to-noise performance. We cite this important work (ref.19) in the revised manuscript for advanced technologies.

2) Towards the end of the introductory paragraph the authors discuss about "Several experimental platforms..." that demonstrate PT-symmetry. Indeed there are electronics,

acoustic, matter waves, spintronics ect. The authors have to give some representation references there.

Reply: We thank the Reviewer #1 for this comment. We now list the possible experimental platforms for the realization of PT symmetry and the corresponding representation references in the revised manuscript.

3) It is probably a "language" slip but at the top of page 3 (second paragraph before the end of introduction) the authors state that the previously used Floquet platforms "require active elements". I do not think that they are "requiring" active elements. But it is true that they have been performed using active elements. I think that it is more correct to say instead "...have used active elements...".

Reply: We thank the Reviewer #1 for this useful suggestion. We change the sentence to "have used active elements" in the revised manuscript, which is a more accurate description.

4) At the section of "Experimental realization of the Floquet PT-symmetric System" there are many typos (most coming from the "ket" notations. Please correct.

Reply: We thank the Reviewer #1 very much for pointing out this issue. There was a mistake with the ket notation and we fix it in the revised manuscript.

5) In the case of the amplified signal, are there any nonlinear saturation effects? If yes, is there an expectation when they kick in?

Reply: We appreciate the Reviewer #1 for posing this intriguing question. In our current study, the waveguide array employed is a passive system, devoid of any nonlinear saturation effects, as the amplification signal is not actually achieved. Moving

forward, we intend to investigate the potential for captivating nonlinear effects by utilizing a genuinely active system in our future research endeavors.

6) Reference 34: There is a missing "PT" in the title of this paper. Also in Ref. 34 the title starts with small letter i.e. "spectroscopy...". Please correct. Also, the authors might want to cite the following theoretical work that also discuss Floquet PT-symmetric systems (although this is not necessary): Wave-packet self-imaging and giant recombinations via stable Bloch-Zener oscillations in photonics lattices with local PT-symmetry, N. Bender, H. Li, F. M. Ellis, T. Kottos, Phys. Rev. A 92, 041803(R) (2015).

Reply: We thank the Reviewer #1 very much for pointing out these mistakes in the references. In the revised manuscript, we correct these mistakes and check the format of the references throughout the manuscript to make sure no further mistakes exist. We cite the interesting work mentioned by the Reviewer #1 in the introduction of PT-symmetry in the revised manuscript.

Provided that the above comments are addressed appropriately, I recommend publication of this paper to Nature Communications.

Reply: We thank the Reviewer #1 again for the careful reading of our work and important comments which improve our manuscript significantly.

Responses to Reviewer #2

This manuscript presents an experimental realisation of an integrated optical waveguide coupler with specially introduced periodically modulated loss, such that the evolution is mathematically equivalent to a Floquet parity-time (PT) symmetry dynamics. While this problem has been extensively investigated theoretically and experimentally in other contexts, the presented realisation with integrated waveguides appears to be novel.

Reply: We thank the Reviewer #2 for their positive assessment of the quality and novelty of our work.

There appear several questions

1) It is stated in lines 93-94 that “The Hamiltonian $H(t)$ describing this Floquet process is PT-symmetric, i.e., $PTH(t)PT=H(t)$ ”. However, this is not generally correct, depending on where the zero of the time (distance) axis is selected. Specifically, only if $t=0$ is in the middle of one segment with loss in either left or right waveguides, then indeed $PTH(t)PT=H(t)$. But if $t=0$ is at the boundary between the “up” and “down” segments, then one actually identifies anti PT symmetry since the time-reversal operator swaps the two segment types, meaning that there appears a minus sign on the RHS as $PTH(t)PT=-H(t)$. Please clarify and add a critical discussion on these aspects.

Reply: We thank the Reviewer #2 for raising this point. We emphasize that the z -axis in our system represents the arrow of time, so flipping z implies flipping time rather than flipping the sign of a real space coordinate. Therefore, reversing the z direction results in the same physical consequence as reversing time. Namely, under $z \rightarrow -z$ ($t \rightarrow -t$), due to the effect of time-reversal operator (which is equivalent to complex conjugate operator in our case), gain becomes loss, loss becomes gain), hence the gain and loss segments swap with each other, which does not depend on the choice of defining the origin $t=0$. Therefore, the Hamiltonian preserves PT-symmetry independent of the choice of $t=0$. We elaborate this point further in the revised manuscript.

2) It would be useful to cite and discuss the following relevant references: Pseudo-Parity-Time Symmetry in Optical Systems <https://doi.org/10.1103/PhysRevLett.110.243902>, Stabilizing non-Hermitian systems by periodic driving <https://doi.org/10.1103/PhysRevA.91.042135>

Reply: We thank the Reviewer #2 for bringing these interesting references to our attention.

The first reference elucidates the implementation of pseudo-PT symmetry by modifying the real component of the refractive index in the material. This approach enables the transformation of a non-PT symmetric system into a PT-symmetric one, thereby facilitating the manipulation of its properties. The second reference delves into the utilization of periodic driving to engender an effective real spectrum in a non-Hermitian system devoid of PT symmetry. We cite these valuable works and integrate a comprehensive discussion pertaining to their relevance to our research in the revised manuscript (Page 2, sentence in red).

Overall, this work can be potentially recommended for publication after a revision.

Reply: We thank the Reviewer #2 again for the potential recommendation for publication in Nature Communications and the very helpful comments and suggestions to improve our work.

Responses to Reviewer #3

In this work, the authors designed an integrated photonic waveguide and explored the Floquet PT-symmetry. They experimentally demonstrated the PT-symmetry breaking transition and achieved the control of amplification and absorption through excitation ports in their setup. Overall, this paper is clearly written. The reported results could make an interesting contribution to the experimental realization of non-Hermitian Floquet systems. However, the PT symmetry, exceptional point and the related non-Hermitian phenomena have been intensively studied in the past decades. In photonic waveguides, the PT symmetry and its breaking have been considered in many theoretical and experimental studies, e.g., see the following research articles and reviews: PHYSICAL REVIEW RESEARCH 3, 023211 (2021); <https://doi.org/10.1002/adma.201903639>; <https://doi.org/10.1002/apxr.202200053>; <https://doi.org/10.1093/nsr/nwy011>; Phys. Rev. Lett. 101, 080402; Physics - Uspekhi 57 (11) 1063 - 1082 (2014); <https://doi.org/10.1364/PRJ.6.000A51>;

Specially, an experimental realization of a non-Hermitian Floquet system was reported before in PHYSICAL REVIEW RESEARCH 3, 023211 (2021). Replacing the role of time by a spatial propagation direction is also a well-developed approach to do Floquet engineering in photonic waveguides. Meanwhile, there are many other theoretical works on PT symmetric and more generally non-Hermitian Floquet systems, which are unfortunately ignored by the authors. Putting together, I think that regarding the impact and novelty, the results reported in this work could not meet the high standards of nature communications. Therefore, I could not recommend publication.

Reply: We thank the Reviewer #3 for reading our manuscript and providing these critical comments. We appreciate the references mentioned by the reviewer. After carefully reading these references, we are convinced that these works do not impact the novelty of our work, nor the operation and fundamental contributions of our work to PT symmetry physics. These references majorly focus on the topological features of Floquet systems with PT symmetry, asymmetric light transport, and the basic proof of PT symmetry. However, our work investigates the interplay between Floquet mechanisms and PT-symmetry phase, and experimentally demonstrate PT-symmetry breaking and the new functionalities enabled in an on-chip photonic device. The model we use here is proposed for the first time both theoretically and experimentally, which leads to brand new effects that are not covered in any previous work, hence enabling promising functionalities for on-chip light manipulation and power control.

For example, in the work by Wu et al. [PRR 3, 023211 (2021)], the authors investigated the interaction between PT symmetry and Floquet lattices, and the consequence to its topological phase. In this sense, the Floquet mechanism comes from the modulation of hopping terms in the SSH lattice, not the modulation of onsite potential which determines PT symmetry of the system. Their PT terms are constant, i.e., the gain and

loss are unchanged as mode propagates forward. In clear contrast, in our case the Floquet modulation varies the PT symmetry condition, since gain and loss components are periodically flipped in time (see Fig. 1 below). As a consequence, our model leads to totally different phenomena compared to previous works. First, we show that periodic flips enable spontaneous PT-symmetry phase breaking at very small gain/loss regimes, which relaxes the requirement of large gain and loss for the investigation of EP and PT-symmetry phase breaking. In addition, by judiciously choosing the modulation period, we demonstrate that our Floquet PT-symmetric system enables efficient control of amplification/attenuation of input signals, offering new opportunities to develop on-chip devices for power and signal control. All these exciting results are totally novel, and not reported in previous works. We believe that our results are critical for understanding the foundation of PT-symmetry physics, and could potentially lead to technological advances for on-chip devices.

We thank the reviewer again for bringing these interesting references to our attention, we cite them appropriately in the revised manuscript.

Fig. 1. Schematic plot of the imaginary part of the on-site potential for our model and previous works. In our model, the Floquet mechanism arises from the modulation of the on-site potential, which governs PT symmetry by periodically switching the gain

and loss components over time. In contrast, the previous Floquet models incorporate modulation in the coupling constant. This distinction between our model and the previous one results in distinct effects and functionalities.

There are some other issues that I suggest the authors to address before they resubmit this paper to any other journals:

(i) Some descriptions and definitions in the paper are not accurate. For example, having balanced gain and loss is not the necessary condition for the appearance of PT-broken phases and exceptional points. Spatially asymmetric/nonreciprocal couplings or even a uniform loss could do the job. The authors' related statements in the abstract are thus inaccurate.

Reply: We agree with the Reviewer #3's comment, our Hamiltonian with PT symmetry is not the only way to enable PT-broken phases and exceptional points. Hamiltonians with nonreciprocal couplings may also achieve the same phenomena. For example, a Hamiltonian with the form $\begin{pmatrix} w & -g \\ g & -w \end{pmatrix}$ contains asymmetric couplings and commutes with anti-PT symmetry, and it has the same spectrum as the one of Hamiltonian $\begin{pmatrix} ig & w \\ w & -ig \end{pmatrix}$ which is PT-symmetric, thus both Hamiltonians can support PT-broken phases and exceptional points. More details about this point can be found in our recent review (Nat. Nanotechnol. 18, 706-720 (2023)). We have carefully examined the text throughout the manuscript to make sure no ambiguity and misleading statements are made.

(ii) The concept of Floquet PT-symmetry should be treated more carefully. For stroboscopic dynamics, the Floquet PT-symmetry should be defined at the level of Floquet operator. In more general situations, one should consider the impact of PT operations on both the effective Hamiltonian and micromotion operator. Defining the

Floquet PT-symmetry simply as the PT symmetry of the instantaneous time-dependent Hamiltonian is questionable.

Reply: We thank the Reviewer #3 for the insightful comments and valuable suggestions. We agree with the Reviewer #3 that the symmetry of a Floquet system can be explored from various perspectives. Our proposed structure distinctly stands out from any previous model, introducing periodic and inverse flip modulation to the imaginary part of the PT-symmetric system for the first time. It obeys PT symmetry at arbitrary instants in time. The complex potentials remain constant for half of the period and invert their positions in the next half. Thus, $H(t)$ clearly obeys PT symmetry at all instants in time, in addition it breaks parity symmetry in z or time direction. This novel structure introduces a range of unprecedented functionalities, such as selective amplification and absorption of input signals, spontaneous symmetry breaking at low gain and loss regimes, and power control contingent upon the phase of the initial states. These functionalities have been reported and experimentally validated for the first time in this work. While our model is novel, we have adopted the term "Floquet PT" to capture its inherent periodic modulation and the unique functionalities it introduces. Additional evidence showcasing the distinctive functionalities of our structures can be found in Supplementary S4. These developments represent a significant advancement in the field of PT symmetry. We remain open to suggestions for a more suitable name for this new model with novel functionalities.

(iii) Some obvious typos should be removed. In the lines 181-189 of the PDF file, there are some strange symbols, making the text unreadable. In Refs. [33] and [34], the PT symbols are missing. The authors are suggested to check their text more carefully.

Reply: We thank Reviewer#3 for pointing out the mistakes in the math symbols and references. We correct these mistakes and check carefully throughout the manuscript to

make sure no other mistakes exist.

REVIEWERS' COMMENTS

Reviewer #1 (Remarks to the Author):

I have reviewed the revisions in the manuscript and also the replies of the authors. The authors have adequately responded to all my comments and made all the necessary corrections. I re-iterate that the experimental implementation in the photonics framework of their scheme and the opportunities that it brings (e.g. low values of gain/loss for breaking PT-symmetry) are novel. I, therefore, recommend publication of the paper to Nature Communications in its current form.

Reviewer #2 (Remarks to the Author):

The authors have responded to previous comments; the manuscript is now recommended for publication.

Reviewer #3 (Remarks to the Author):

In the revised manuscript, the authors have addressed most of my concerns. Besides, I find that the statement like "A Floquet PT-symmetric system is formed by periodically swapping in time the location of loss and gain (Fig. 1b), while preserving PT-symmetry at each time instant" is still very misleading and system-specific. As the authors also agree, adding gain and loss is only one physical way to realize a PT symmetric system, and there are many other possibilities. The authors must clearly emphasize that the concept of Floquet PT symmetry they used in their manuscript follows their own definition, and there are other possible definitions that could be valid in other and more general situations in the literature. This work might be considered for publication in NC after the aforementioned point is fixed by the authors.

Response Letter

Responses to Reviewer #1

I have reviewed the revisions in the manuscript and also the replies of the authors. The authors have adequately responded to all my comments and made all the necessary corrections. I re-iterate that the experimental implementation in the photonics framework of their scheme and the opportunities that it brings (e.g. low values of gain/loss for breaking PT-symmetry) are novel. I, therefore, recommend publication of the paper to Nature Communications in its current form.

Reply: We thank Reviewer #1 for recommending our paper for publication in Nature Communications.

Responses to Reviewer #2

The authors have responded to previous comments; the manuscript is now recommended for publication.

Reply: We thank Reviewer #2 for recommending our paper for publication in Nature Communications.

Responses to Reviewer #3

In the revised manuscript, the authors have addressed most of my concerns. Besides, I find that the statement like “A Floquet PT-symmetric system is formed by periodically swapping in time the location of loss and gain (Fig. 1b), while preserving PT-symmetry at each time instant” is still very misleading and system-specific. As the authors also agree, adding gain and loss is only one physical way to realize a PT symmetric system,

and there are many other possibilities. The authors must clearly emphasize that the concept of Floquet PT symmetry they used in their manuscript follows their own definition, and there are other possible definitions that could be valid in other and more general situations in the literature. This work might be considered for publication in NC after the aforementioned point is fixed by the authors.

Reply: We thank Reviewer #3 for the comments. Following this suggestion, in the updated manuscript we have emphasized the approach of realizing PT symmetry employed in our work and highlighted our specific definition of Floquet PT symmetry (marked in red color), while admittedly other possible definitions may exist in the literature. Finally, we thank again Reviewer #3 for carefully reviewing our manuscript and for the potential recommendation for publication in *Nature Communications*.